

# Geodynamic diagnostics, scientific visualisation and StagLab 3.0

Fabio Crameri

Centre for Earth Dynamics and Evolution (CEED), University of Oslo, Postbox 1028 Blindern, 0315 Oslo, Norway

*Correspondence to:* Fabio Crameri (fabio.crameri@geo.uio.no)

**Abstract.** Today's Geodynamic models can, often do, and sometimes have to become very complex. Their underlying, increasingly elaborate numerical codes produce a growing amount of raw data. Post-processing such data becomes therefore more and more challenging and time consuming. In addition, visualising processed data and results has, in times of coloured figures and a wealth of half-scientific software, become one of the weakest pillars of science, widely mistreated and ignored. Efficient and automated Geodynamic diagnostics and sensible, scientific visualisation, preventing common pitfalls, is thus more important than ever. Here, a collection of numerous diagnostics for plate tectonics and mantle dynamics is provided and a case for truly scientific visualisation is made. Amongst other diagnostics are a most accurate and robust plate-boundary identification, slab-polarity recognition, plate-bending derivation, surface-topography component splitting and mantle-plume detection. Thanks to powerful image processing tools and other elaborate algorithms, these and many other insightful diagnostics are conveniently derived from only a subset of the most basic parameter fields. A brand-new set of scientifically proof, perceptually uniform colour maps including '*devon*', '*davos*', '*oslo*' and '*broc*' is introduced and made freely available. These novel colour maps bring a significant advantage over misleading, non-scientific colour maps like $'rainbow'$, which is shown to introduce a visual error to the underlying data of up to 7.5 %. Finally, STAGLAB (www.fabiocrameri.ch/software) is introduced, a software package that incorporates the whole suite of automated Geodynamic diagnostics and, on top of that, applies state-of-the-art, scientific visualisation to produce publication-ready figures and movies, all in a blink of an eye, all fully reproducible. STAGLAB, a simple, flexible, efficient and reliable tool, made freely available to everyone, is written in MatLab and adjustable for use with Geodynamic mantle-convection codes.



# Contents





# 1 Overview

The first, basic numerical Geodynamic models were developed in the early seventies (e.g., Minear and Toksöz, 1970; Torrance and Turcotte, 1971). Since then they have become more powerful and often more complex (see e.g., King, 2001; Gerya, 2011; Lowman, 2011; Coltice et al., 2017). Indeed, dynamically self-consistent Geodynamic models used to reproduce the first-order characteristics of the complex plate-mantle system, like mobile surface plates, single-sided subduction and mantle plumes, need a certain complexity (e.g., Crameri and Tackley, 2015). However, this complexity often inhibits a simple understanding of the full interplay between all individual physical aspects of these models: The models become too complicated to be easily explained or even fully understood. In addition, more-elaborate numerical codes powering these models (e.g., Zhong et al., 2000; Gerya and Yuen, 2007; Moresi et al., 2007; Tackley, 2008; Davies et al., 2011; Thieulot, 2014; Kaus et al., 2016; Heister et al., 2017) and the still increasing computational power available for their execution produce more and more raw data. The resulting amount of data to be processed easily exceeds the capability of a human scientist. Today, efficient, automated and intelligent Geodynamic diagnostics are thus more important than ever to keep up with the advances made in numerical models.

Moreover, conveying new findings to the community critically depends on data visualisation as figures are pivotal to make raw data tangible, understandable and explainable (e.g., Gerya, 2010). However, it has become increasingly difficult to visualise research given the increasing complexity of models (e.g., high-resolution and 3-D geometry) and new and improved visualisation techniques (e.g., coloured figures and movies). Worrisome visualisation pitfalls arise that make figures confusing, unreadable, or even misleading and hence unscientific. The *rainbow* colour map strongly deteriorates, for example, the

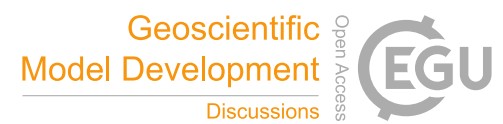

underlying scientific data (Pizer and Zimmerman, 1983; Ware, 1988; Tufte, 1997) due to the inhomogeneous colour sensitivity of the human retina (Thomson and Wright, 1947). On top of that, it is confusing to people with some of the most common colour-vision deficiencies. Even though all this is known since a while (see Rogowitz and Treinish, 1996, 1998; Light and Bartlein, 2004; Borland and Ii, 2007, and $\#endrainbow$), the *rainbow* colour map is still commonly used by scientists and

commonly accepted by scientific journals. In addition to the current need for more-advanced Geodynamic diagnostics, it is thus clear that, today, efficient scientific-proof visualisation has also become more important than ever.

Nevertheless, there is currently no tool available to perform key Geodynamic diagnostics and scientific visualisation efficiently and reliably. STAGLAB (www.fabiocrameri.ch/software) is the post-processing and visualisation software that aims to fill this gap and fulfil all necessities mentioned above. Previous versions of STAGLAB (Crameri, 2013, 2017) were designed

to specifically handle the raw data produced by the finite-difference/finite-volume multi-grid mantle convection code StagYY (Tackley, 2008). STAGLAB 3.0 offers, however, potential compatibility to other Geodynamic codes as well and has, for example, already been tested successfully with output from the widely used, finite-element code Fluidity (Davies et al., 2011).

In the following, I will present key Geodynamic diagnostics and how to automate them, discuss scientific and non-scientific colour schemes and visualisation approaches, and introduce STAGLAB, the all-in-one software package that makes powerful

Geodynamic diagnostics and state-of-the-art scientific visualisation easily available to everyone.

## 2  Geodynamic diagnostics

Here, I list a variety of key diagnostics for mantle convection and plate tectonics that can be automised and applied to digital data. While some of these diagnostics are known, others are significantly improved or newly introduced here (see Table 2 for an overview). The Geodynamic diagnostics covered here are explained in detail below and can be predominantly applied to 2-D

data or vertical slices of 3-D data. All of the diagnostics are implemented in, tested and made ready for use with the software STAGLAB 3.0 (see Sect. 4).

### 2.1  Plate-tectonics diagnostics

A first suite of diagnostics focuses on plate tectonics that operates at, or close to, a planet's rocky surface. A variety of simple, general diagnostics, such as mean plate thickness, maximum plate stress and strain-rate, and upper-mantle temperature, density

and viscosity, are extracted from only a few key parameter fields. The major, more complex diagnostics are outlined below and focus on plate, plate-boundary and slab dynamics as well as long-wavelength surface topography.

### 2.1.1  Regional subduction topography

The topography above a subduction zone typically displays a suite of characteristic regional features. These regional topographic characteristics can be individually measured using automated diagnostic algorithms as explained in Fig. 2 and Table 1.

Specific algorithms are available for the following regional characteristics:





1. **The viscous fore-bulge (or outer rise)**, which is the upward deflection of the subducting plate outboard of the subduction trench. This transient, viscous uplift is mainly caused and controlled by the downward bending of the plate at the subduction zone into the low-viscosity mantle (de Bremaecker, 1977; Crameri et al., 2017). Here, the fore-bulge height, $FB$, is defined as the difference between its maximum elevation, $FB_0$, and the minimum plate-surface elevation, $z_2$, at its side away from the subduction trench (see Fig. 2).

2. **The subduction trench**, which is the downward deflection that is located at the plate boundary precisely indicating the interface at Earth's surface between upper and lower plate. Studies like Zhong and Gurnis (1994) and Crameri et al. (2017) suggest that it is likely of dynamic origin and continuously controlled by a multitude of factors. Here, the trench depth, $TR$, is defined by the maximum depth of the depression ($TR_0$) relative to the model's sea level ($z_1 = 0$ km).

3. **The island arc (or volcanic arc)**, which is the collisional high caused by horizontal plate compression. The plate strength has a major control on its resulting elevation (Crameri et al., 2017) and it is, apart from its dynamic origin, often strongly affected by volcanism (Karig, 1971). Here, the island-arc height, $IA$, here defined by its maximum elevation ($IA_0$) relative to the characteristic upper-plate elevation, $z_4$, outboard the arc.

4. **The back-arc depression (or basin)**, which is the upper-plate depression following the island arc further away from the trench. The depression's origin is often likely two-fold and a combination of upper-plate extension or even spreading (Karig, 1971) and dynamic coupling (via the mantle wedge) with the sinking slab below (Crameri et al., 2017). Here, the back-arc depression depth, $BAD$, is defined as the difference of the maximum depression on the upper plate, $BAD_0$, with the characteristic undeflected upper-plate elevation, $z_4$, outboard the back arc. This is necessary due to variable, isostatically induced elevation differences between upper- and lower plate.

Additional diagnostics of regional surface topography at subduction zones can be measured. The maximum **horizontal extent of the back-arc depression**, $BAD_{extent}$, can, for example, be defined by the distance between trench ($x_5$) and the far end of the back-arc deflection, $x_3$, away from the trench. The latter point can be approximated by scanning the deflection for the one point that is still lower than twice the maximum vertical variation occurring in the undeflected reference upper-plate portion. Another diagnostics is given by the **volume of the back-arc depression**, which on a 2-D plane corresponds to a vertical area. It can be approximated by the area constructed by the maximum basin horizontal ($x_3 - x_4$) and vertical extent ($z_4 - BAD_0$). Yet another useful diagnostics is the **tilt of the upper plate** towards the subduction trench as it has been shown to vary significantly during subduction evolution (Crameri and Lithgow-Bertelloni, 2017). The tilt can be tracked by a measure taken at a certain critical distance away from the trench. The tilt angle measurement is significantly improved (i.e., made more robust and less fluctuating) by taking a mean of multiple measurements taken just next to each other.

It is generally good practice to normalise the vertical amplitude of the characteristic topographic points mentioned above to a characteristic plate thickness defined, for example, by the thermal lithosphere thickness. This allows scaling obtained results to systems with different Rayleigh numbers and hence different plate thicknesses.





### 2.1.2 Topography components

In addition to the absolute surface topography, its *isostatic* and the remaining, non-isostatic *residual* components are useful to understand the various and diverse sources of long-wavelength surface elevation. To derive these two topography components, the plate thickness, $d_p$, has to be tracked along the horizontal extent of the model (see Sect. 2.1.3). As introduced in Crameri

et al. (2017), the **isostatic topography component** for each vertical column along the model extent can be calculated using the base of the plate (e.g., as defined by a 1700 K isotherm) as compensation depth. Depending on whether the plate is denser or lighter than the mantle, it is then given by

$$z_{topo,iso}(\rho_p) = \begin{cases} \frac{(\rho_m - \rho_p)d_{LAB}}{\rho_p - \rho_{air}}, & \text{if } \rho_p \leq \rho_m \\ \frac{(\rho_m - \rho_p)d_p}{\rho_m - \rho_{air}}, & \text{if } \rho_p > \rho_m \end{cases} \tag{1}$$

with $\rho_p$ as the vertically averaged plate density at each horizontal point in space, $\rho_m$ the horizontal mean upper-mantle

density just below the plate and away from any sinking slab, $\rho_{air}$ the air density, $d_{LAB}$ the variable thermal lithosphere–asthenosphere boundary (LAB) depth as defined, for example, by a 1700 K isotherm along the model, and $d_p$ the variable plate thickness at each horizontal point in space that includes surface topography, so is the thickness between the rock–air interface and the base of the plate.

Crucially, the resulting isostatic topography component has to be normalised throughout the model to produce a mean

topography that corresponds to the sea level according to

$$z_{topo,iso,0}(x) = z_{topo,iso}(x) - \langle z_{topo,iso} \rangle \tag{2}$$

with $\langle z_{topo,iso} \rangle$ as the mean of the model-wide isostatic topography component and $x$ as the horizontal coordinate. This therefore ensures accounting for both the conservation of volume in an incompressible mantle and the coherence of the surface plate.

The **residual topography component** corresponds then simply the non-isostatic part of the topography and is given by

$$z_{topo,res} = z_{topo,total} - z_{topo,iso,0} \tag{3}$$

with $z_{topo,total}$ as the surface topography of the model. The residual topography component can be considered as the part of topography that cannot be explained by the plate's isostatic dynamics. It is important to point out that it can, however, be caused not only by dynamic sources from within the convecting mantle below, but also from sources (e.g., horizontal tectonic

forces) within the plate itself.

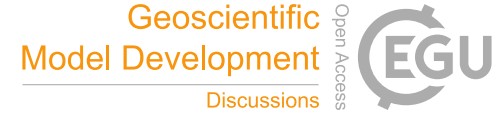



### 2.1.3  Plate thickness and plate-core depth

An estimate of the plate thickness can be derived using temperature isotherms (generally the 1600 K isotherm) according to the *thermal*-lithosphere definition. A maximum depth limits the derived plate thickness at subduction zones, where the plate and hence the isotherms bend downward into the mantle. By definition, this approach simultaneously allows to track the
lithosphere–asthenosphere boundary and its topography.

Another, more widely applicable option is to derive the plate thickness automatically from individual radial profiles of temperature (or viscosity). Since a typical plate cools via diffusion, the local temperature profile is (nearly) linearly decreasing throughout the boundary layer (from top to bottom) and becomes (nearly) isothermal in the convecting mantle below. The transition from diffusion-dominated to the convection-dominated cooling is marked by a characteristic sharp bend in the profile.
The bend in the profile can be tracked automatically by scanning for it from top to bottom. This gives an estimate of the local plate thickness, or a **mean plate thickness**, when performed on a horizontal mean (i.e., root-mean-square) temperature profile. The **plate-core depth** lies then simply about half-way down between the surface and the plate base. This depth is particularly useful to robustly track plate velocities and other similar characteristics.

### 2.1.4  Stagnant-lid diagnostics

Checking for a stagnant lid (i.e., the absence of subduction) can be done using a combination of tests. Key indicators for a stagnant lid are a low overall plate-thickness variation, a low maximum plate thickness throughout the model domain (i.e., no incipient subduction zone), the lack of cold lithosphere in the upper mantle (i.e., no mature subduction zone), and a low relative motion within the surface plate (a stagnant lid is mostly rigid).

If a stagnant lid is present, various diagnostics can be performed. Stagnant-lid diagnostics include, for example, the maximum
plate stress, the maximum depth of plastic failure (i.e., brittle or ductile yielding), and the lithosphere–asthenosphere boundary topography. The two latter physical complexities have been shown to be a reliable indicator for subduction initiation (Crameri and Tackley, 2016).

### 2.1.5  Plate-boundary tracking

Using more advanced routines, converging and diverging plate boundaries can be tracked robustly and fully automatically.
A basic, first-order plate-boundary tracking can already be achieved by diagnosing temperature and velocity fields only. A plate-boundary tracking routine can, however, be improved when additional parameter fields are considered (see Table 3).

To robustly find converging and diverging plate boundaries (i.e., **subduction trenches** and **spreading ridges**) within a 2-D model at a given point in time, a quite elaborated procedure is necessary (see Fig. 1). Several checks need to be performed initially to exclude the presence of a stagnant lid, and thus the absence of subduction or spreading zones (see Sect. 2.1.4). If a
stagnant lid is ruled out, potential plate boundaries can be located. Finding plate boundaries robustly necessitates finding all, sharp and diffuse plate boundaries, finding the exact location at the very plate surface (e.g., the outcropping of the subduction fault) while accounting for plate topography, and preventing multiple tracking of one and the same plate boundary.





The first derivative of the plate velocity indicates both convergent and divergent plate boundaries quite robustly by clearly detectable peaks. However, plate boundaries can sometimes become quite diffuse. Spreading ridges in coarsely discretised numerical models are, for example, often diffuse due to various secondary ridges forming in the young and weak plate around the main ridge. To find both sharp boundaries accurately and not miss diffuse boundaries, critical values for the velocity change

across plates and the horizontal distance between the measurement points need to be quite conservative (i.e., a low velocity change measured far apart from the boundary). To not loose accuracy on the location, best practise is to start with a conservative check and gradually make critical values more restrictive to the point shortly before the boundary is not detected any longer.

The plate velocity at different depth levels inside the plate (e.g., at the plate surface and plate core; see Sect. 2.1.3) can be used to distinguish between shallow and deep plate boundaries, and their horizontal offset additionally serves additionally as

an indicator for the polarity of the subduction system.

### 2.1.6   Plate velocities

Knowing the location of plate boundaries, the velocities of the converging plates can be diagnosed. The individual plate velocities are best measured in the cold core of the plates (see Sect. 2.1.3), and close, but not too close to the trench. Once the subduction polarity is found (see Sect. 2.1.8), upper- and lower-plate can be distinguished, and the **trench velocity** can

be measured simply by the velocity of the upper plate just next to the trench. For comparison to the effective trench retreat, a **theoretical trench velocity** can be calculated during any given subduction phase. Assuming a non-deformable slab (which is not always the case), the current trench velocity can be approximated by

$$v_{TR,theoretic} = \frac{v_{Stokes}}{\tan\theta} \tag{4}$$

where $v_{Stokes}$ is the vertical sinking velocity of the slab and $\theta$ is the shallow-depth slab dip angle (e.g., Capitanio et al.,

20    2007).

### 2.1.7   Plate age

A **theoretical subducting-plate age**, $a_{Plate,theoretic}$, at the subduction trench can be derived making use of the theory of the half-space cooling model as

$$a_{Plate,theoretic} = \frac{\left(\frac{h_{LP}}{2.32}\right)^2}{\kappa} \tag{5}$$

where $h_{LP}$ is the thickness of the subducting plate at the trench and $\kappa$ is the thermal diffusivity (Turcotte and Schubert, 2014).





### 2.1.8 Slab-dynamics diagnostics

The current **slab-tip depth** can simply be tracked using the deepest point in a temperature contour outlining the cold material of the sinking plate. It can be refined by taking not the deepest point within the contour, but the point inside the contour that is the farthest away from the trench, as the slab tip usually is.

The **subduction polarity** of an asymmetric subduction zone can be found by checking for cold material in the uppermost mantle at two depth levels. A subduction polarity should only be given, if the coldest spots at these two depth levels are shifted horizontally with respect to one another over a certain threshold.

If the subduction polarity is found, the **shallow-depth slab dip**, $\theta$, can additionally be measured between these two depth levels, $z_{5A} = 1.25 d_p$ and $z_{5B} = 7/4 z_2$, where $d_p$ is the mean lithosphere thickness away from the subduction zone. This results
in a measuring depth, $z_5$, for the slab-dip angle located just below the lithosphere (see Fig. 2) according to

$$z_5 = z_{5A} + \frac{z_{5B} - z_{5A}}{2}. \tag{6}$$

The minimum, maximum and mean **slab viscosity** can be automatically derived using a specified area inside the slab. This area can be set in such a way that it spans the points outlining the sinking plate that lie within a square box around the point of maximal slab bending (see Sect. 2.1.9). If, for some reason, this method fails, the slab viscosity can instead be derived by
spotting the coldest portion inside a sinking slab at a depth level below the surface plate. There, a small region surrounding the coldest spot is checked for the mean, the minimum and maximum viscosity. Depending on the task, either the mean value (e.g., for the indication of absolute slab viscosity) or the minimum value is more useful (e.g., for the calculation of bending dissipation; see Sect. 2.1.10).

The **slab-mantle viscosity difference** can be derived from the minimum viscosity found in the slab (as outlined above)
and the viscosity found in the surrounding upper-mantle. The **upper-mantle viscosity** can be approximated by the horizontal median of viscosity found in the upper mantle below the plate. As such the anomalously high viscosity of the few cold regions (i.e., the sinking slabs) are weighted much less than the viscosity of the more common hot mantle surrounding them. The slab-mantle viscosity difference, $\Delta\eta_{LA}$, is then the difference between slab- and mantle viscosity.

The **slab sinking rate**, $v_{Slab}$, is simply extracted using the vertical velocity measured at $z_{5B}$. An approximation of the total
amount of water transported to the mantle can be calculated using the scaling law of Magni et al. (2014). The theoretical **total water retention**, $W$ [kg/m$^2$], of the slab is then

$$W = (1.06 v_{Slab} + 0.14 a_{Slab} - 0.023 T_{Mantle} + 17) 10^5 \tag{7}$$

where $v_{Slab}$ [cm/a] is the sinking velocity of the slab, $a_{Slab} = a_{Plate,theoretic}$ [Ma] is the age of the slab, and $T_{Mantle}$ [°C] is the potential mantle temperature.



### 2.1.9 Plate bending

The **subduction bending radius**, $R_B$, can be fully automatically calculated using the current plate geometry. A spline method with temperature contours can be used to outline the subducting-plate shape (see Petersen et al., 2016, for other methods). A certain temperature threshold thereby extracts cold plate portions including a sinking slab if present. Extracting the lowermost

points in every horizontal column along the numerical grid then marks the down-going plate and removes simultaneously any non-subducting plate portions. The resulting band of points needs then to be fitted using a smoothing spline. This provides a line to represents the down-going plate geometry. The local curvature, $R_{curv}$, along the line over its whole width is given by

$$R_{curv}(x) = \frac{\left[1 + \left(\frac{dz}{dx}\right)^2\right]^{3/2}}{\left|\frac{d^2z}{dx^2}\right|} \tag{8}$$

with $dx$ and $dy$ as the incremental spatial change of the line at the horizontal location, $x$, in x- and z-direction, respec-

tively. The minimum plate-bending radius at the kink of the subduction zone, $R_B$ (see Fig. 2), corresponds then simply to the maximum bending found along the considered plate portion and is thus

$$R_B = min\left[R_{curv}\right]. \tag{9}$$

### 2.1.10 Viscous bending dissipation

The **viscous bending dissipation** within the lithosphere at a subduction zone can be calculated using the above definition of

the subduction bending radius. Conrad and Hager (1999) provides an approximation for the viscous bending dissipation for the case of a purely viscous plate. Including the additions outlined in Buffett (2006) to consider a visco-plastic plate, and neglecting dissipation in the subduction channel, the viscous dissipation is finally given by

$$\phi_L^{vd} = C_l v_p \sigma_{y,p} \left(\frac{d_s^2}{R_B}\right) \tag{10}$$

with $C_l = 1/6$ as a constant, $v_p$ as the lower-plate velocity, $d_s$ the slab thickness, and $\sigma_{y,p}$ as the maximum yield stress

within the bending portion of the plate. The lithospheric bending dissipation, $\phi_L^{vd}$, can be normalised using a certain value, $\phi_{L,char}^{vd}$ in $\mathrm{W\,m^{-1}}$, to a normed value

$$\phi_{L,norm}^{vd} = \frac{\phi_L^{vd}}{\phi_{L,char}^{vd}}. \tag{11}$$

### 2.2 Mantle-dynamics diagnostics

This second suite of diagnostics focusses on the mantle dynamics operating in a planet's interior. The major diagnostics are

outlined below, and focus on the residual mantle temperature and active and passive up- and downwellings including mantle plumes.



### 2.2.1 Residual mantle temperature

Extracting the residual temperature in a given domain is often very insightful, as Geodynamic flows are often strongly temperature dependent and hence driven by the temperature anomalies. The residual mantle temperature can be defined in different ways. Most-commonly, residual mantle temperature is defined as the temperature anomaly after normalising the temperature at each depth to the corresponding global, horizontal mean (e.g., Labrosse, 2002). This definition is, however, less useful to distinguish local anomalies in global, wide aspect-ratio models. To distinguish a local anomaly, the field has to be normed to a regional instead of a global mean. Therefore there are different ways of defining a residual mantle temperature:

1. **Horizontal Residual:** The field is normalised to the horizontal mean at each depth.

2. **Global Residual:** The field is normalised to the global mean.

3. **Horizontal-Band Residual:** The field is normalised to the mean across a finitely thick, horizontal band at each depth.

4. **Regional Residual:** The field is normalised to the regional mean surrounding each point in space.

The different definitions of mantle temperature anomalies are further used to diagnose mantle diagnostics like up- and downwellings.

### 2.2.2 Up- and downwellings

As highlighted in Fig. 3b, diagnosing thermally **passive and active up- and downwellings** is useful to distinguish whether material in a certain region is rising or sinking. Moreover, information can be extracted about whether this regional flow is thermally self-driven (i.e., active) or induced (i.e., passive). This task can be achieved using the actual flow field (i.e., velocity) and one kind of regional mantle residual temperature (see Sect. 2.2.1). The latter provides information, whether a patch of material is more or less buoyant than its direct surrounding.

### 2.2.3 Mantle-plume tracking

**Mantle plumes** can be tracked using the information outlined above about active up- and downwellings and a plume-tracking algorithm based on the one described in Labrosse (2002). Mantle plumes are here defined (and tracked) as hot or cold up- or downwellings that emerge and are connected to either of the two (hot or cold) boundary layers of the convecting flow (i.e., mantle). Hot and cold temperatures are marked as anomalies when the temperature at a given location exceeds a certain threshold ($f_{hot}$ or $f_{cold}$) in the range between the horizontally averaged temperature ($T_{mean}$) and the maximum ($T_{max}$) or minimum ($T_{min}$) temperature at a given depth level ($z$) according to

$$T_{A,hot}(z) = T_{mean}(z) + f_{hot}[T_{max}(z) - T_{mean}(z)] \tag{12}$$





$$T_{A,cold}(z) = T_{mean}(z) + f_{cold}[T_{min}(z) - T_{mean}(z)] \qquad (13)$$

where for hot anomalies, a threshold of $f_{hot} = 1$ (or $f_{hot} = 0$) defines anomalies that are 100 % (or 0 %) hotter than the horizontal average in the possible range between the mean and the maximum temperature. Once all the anomalies are located, they are checked on their connection to their respective boundary layer by a classical image processing procedure, searching for connected pixels in a matrix (e.g., Kovesi, 2000). The hot and cold thresholds can thereby be chosen separately.

## 2.3 Field-variation diagnostics

Histogram plots of the variation of a specific field along a specified horizontal surface can proof very useful to get insight into the statistical physical behaviour of state of a Geodynamic system like the Earth's mantle (e.g., Supplementary Fig. S1). These statistics can provide mean and median values as well as the standard deviation of the parameter field under consideration. This kind of diagnostics can not only be applied to a perfectly horizontal surface, but also along a vertically slightly variable surface like, for example, along the surface-plate core. This enables, for example, the diagnostics of the strain-rate distribution within the surface plate(s).

## 3 Scientific visualisation

There is not one single right way of visualising scientific data, but there are certain important ways to improve the presentation of scientific data and – crucially – there are several severe pitfalls that have to be prevented when doing so (e.g., Rougier et al., 2014). Here, I outline the unpleasant implications when using a non-scientific colour map (like $'rainbow'$), introduce a novel set of reliable, scientifically tested colour maps, and outline additional ways to improve the positive impact of scientific figures.

### 3.1 Colours

Displaying and even printing scientific figures in colour has become standard for most journals. Random or even physically based colour schemes that disregard the human eye's uneven colour perception (or its most-common mutations) have most-likely multiple serious drawbacks. Therefore, they should be fully avoided by the science community (including authors and journals). Unfortunately, such colour maps are still widely and frequently used, even in high-impact scientific publications. I will therefore outline the most severe problems of unscientific colour schemes below and provide a novel set of ready-to-use, scientifically proof alternatives.

#### 3.1.1 Unscientific colour schemes

Using fancy colour schemes incorporating the whole colour range is appealing: they look peppy and have a lively look with their varying contrasts and multiple colours. Additionally, their main representative, the *rainbow* (a.k.a. *jet*) colour map was,





and in some visualisation programs still is, the default. These unscientific colour maps are thus widely blindly applied by authors while rarely criticised by reviewers and editors.

A colour scheme is unscientific as soon as it features one of the following aspects:

1. **Both red and green colours:** Various forms of colour deficiencies can exist in human eyes, some of which make, for example, green and red undistinguishable.

2. **Multiple different colours with similar lightness:** Colours like red, green and blue with similar lightness cannot be readily ordered to one another (e.g., from low to high values).

3. **No gradual lightness gradient:** A lack of a constant lightness gradient (from light to dark or vice versa) makes a colour map unreadable when printed in black and white.

4. **No perceptual uniformity:** Perceptually non-uniform colour maps cause different parts of the data to be weighted differently (see Fig. 5). The green–cyan part of the colour spectrum has a lower contrast to the human eye than the yellow–red part. The greenish colours hide therefore low-amplitude data variation compared to reddish colours that amplify them.

Whether colours make a figure confusing or even unreadable to colourblind people or whether colours introduce dramatic visual artefacts (see e.g., Light and Bartlein, 2004; Borland and Ii, 2007, and Fig. 5), the figure surely cannot be considered suitable for science any longer. In fact, applying the most moderate form of the commonly used *rainbow* colour map introduces an estimated error (calculated from the change of CIE76 lightness along the colour map) to the represented, underlying data of up to the staggering amount of 7.5 % (see Fig. 5).

The mislead widespread view that visualisation is not an important part of science and thus also not worth spending time and money on (e.g., for external visualisation expertise) is therefore fundamentally wrong.

### 3.1.2 Scientific colour schemes

Useful and clear guidelines on what colour schemes to use and how to judge a given colormap have already been provided elsewhere in detail (e.g., Healey, 1996; Kelleher and Wagener, 2011; Silva et al., 2011). The most important points for choosing a suitable colour scheme can be summarised as follows:

1. **Perceptual order:** The different colours of a colorbar should be perceived as having the same order as the represented numerical values. A temperature scale should, for example, be represented by using the notions of cold and warm colours (and their proportional mixtures).

2. **Uniformity and representative distance:** Two colours should convey the distance of numerical value between them, and colours representing equally differing values should also seem equally different. Clearly separated values must additionally be represented by clearly distinguishable colours, and closer values must be represented by colours perceived to be closer.



3. **No artificial boundaries:** If there are no boundaries in the represented numerical values, the colour scale should not create boundaries, but should rather look continuous.

4. **Separation of bivariate information:** Two (or more) parameter fields represented in the same figure should be clearly separated by two clearly different colour schemes with no repeated colours.

A suitable, scientific colour map makes a figure more intuitive and easier to understand and does not distort the underlying data. This can be done by adjusting a colormap to the parameter's nature and/or to the kind of parameter visualisation. Adjustment to a parameter's nature might be to plot temperature with a blue to red colour scheme as it is intuitively linked to a human's conception of hot (red) and cold (blue) as mentioned above. Adjustment to the kind of visualisation means, for example, that low to high values might be represented by varying the colours from white (low) to red (high), while a variation

between two similar colours (e.g., blue and green) might clarify displaying positive and negative values around a given zero level (e.g., white).

Here, I introduce a novel set of fully scientific, perceptually uniform colour schemes. Crucially, these intuitive colour schemes do not distort the data; they are readable even by people with a colour-vision deficiency; they are readable after being printed in black and white; they add no significant error to the underlying data; the scientific diagnosis of each indi-

vidual colour map is provided alongside the colour map; they are available in all of the most common data formats; they are freely available. Included in the perceptually uniform and visually appealing suite of novel colour maps shown in Fig. 4 are '*devon*', '*davos*', '*oslo*', '*bilbao*', '*lajolla*', and a perceptually uniform version of the common '*gray*' colour map, named '*grayC*'. Three additionally included colour maps '*broc*', '*cork*' and '*vik*' are zero-centred and hence particularly suited for bipolar plots. All of these colormaps perform significantly better in scientific tests for, for example, uniform perception and

local variations in colour contrast (Kovesi, 2015). The full suite of these novel perceptually uniform colour maps including their individual scientific test results is freely available from www.fabiocrameri.ch/visualisation).

### 3.2   Figure design

Good scientific visualisation is accurate and clear. Plots need to show all, and only, the relevant aspects like the relevant data itself, axis ticks and labelling, clear colorbars and instructive titles. Removing unnecessary clutter like for example duplicated

labelling or unnecessary graphical forms, choosing a clear sans-serif font and a lighter (e.g., grey) colouring of axis labels does shift the visual focus on the most important part of the plot: the data.

A visual focus is especially useful when plotting multiple subplots next to each other. Subtle visual guides (that for example group subplots or relate them to each other) can additionally help the reader to understand the figure and by improving its clarity. Magnifying panels can be added to provide a view on the details while not loosing the grand picture. Adjusting the

background colour is useful to adjust the figure to fit seamlessly into its surrounding whether this is a white conference poster or a dark presentation slide. Such visual refinements improve scientific figures and allow them to convey the precious scientific data accurately and be easily understandable and visually appealing.





## 4 StagLab: The software

STAGLAB is the all-in-one, easy-to-use software that combines all Geodynamic diagnostics (see Sect. 2) and the state-of-the-art scientific visualisation (see Sect. 3) outlined above and makes it accessible to everyone. Here, I provide an overview on various aspects of the software, including some of the powerful features, its thoughtful code design, as well as the helpful
external contributions to it, and outline how easily it can be used.

### 4.1 Supported model data

STAGLAB is optimised for the Geodynamic finite-difference code StagYY (Tackley, 2008) but is easily made compatible to other codes. STAGLAB has, for example, already been used with 2-D output from the finite-element code Fluidity (Davies et al., 2011) that, to the contrast, employs an unstructured numerical discretisation (see Supplementary Fig. S2).
The data input for parameter fields simply has to be adjusted according to the input routine $f\_readOther$.

### 4.2 StagLab features

The STAGLAB user receives constant support from a built-in, friendly artificially-intelligent operator, **fAIo**. It facilitates finding input data and saving figures and movies, prevents unnecessary errors, ensures unbroken forward-compatibility of parameter files and keeps STAGLAB itself up-to-date. In the following, I list the key features that lie at the feet of such a streamlined user
experience that make STAGLAB truly boost existing Geodynamic models and the research behind them.

#### 4.2.1 Dimensional scaling

For performance reasons, numerical models are often run using non-dimensional numbers. If a Geodynamic code can be run in both dimensional and non-dimensional mode, STAGLAB will account for that by checking if the data files are dimensional or not, and adjusts the dimensions fully automatically. Moreover, STAGLAB offers the possibility to convert non-dimensional
values into sensible dimensional numbers, using its **Dimensional Mode** (`SWITCH.DimensionalMode`). A given set of dimensionalisation parameters can be defined in the function file $f\_Dimensions$ and then referred to by setting the corresponding flag to the variable `IN.Parameter` in the parameter files.

#### 4.2.2 Automated Geodynamic diagnostics

STAGLAB's incredible diagnostic capabilities (see Sect. 2) decipher Geodynamic models within a fleetingly brief time span. In
fact, performing the whole suite of plate-tectonics diagnostics listed in Table 2 with STAGLAB 3.0 within less than *2.2 seconds* for a high-resolution ($512 \times 256$) 2-D model on a power-efficient laptop (1.3 GHz processor; 8 GB RAM) is record breaking if not revolutionary. During this fleetingly short time period, real-time output is provided listing the most-important model details and diagnostics, and in case of problems, instructive warnings and error messages.



The real power of STAGLAB's Geodynamic diagnostics lies, however, not only in its speed itself, but rather in the combination of both speed and robustness. Given the enormous variety occurring in a Geodynamic system and related numerical models, STAGLAB's diagnostic routines have been trained excessively to become incredibly robust (see e.g., Figure 6).

### 4.2.3  Plot and figure design

5  STAGLAB determines the model geometry automatically from the structure of the input data. It can handle **2-D Cartesian** and **2-D cylindrical** geometries with different aspect ratios as well as **3-D Cartesian** and even **3-D spherical** geometries. The 3-D data can be represented both with *2-D slices* (in any direction normal to a side boundary) or with *3-D isosurfaces*. For Cartesian models these two methods can even be combined into one figure. For 3-D spherical model data, STAGLAB additionally offers a large variety of *map projections*.

10  STAGLAB's visualisation routine is trimmed for accuracy, clarity and simplicity. Plots produced with STAGLAB show all the relevant data like axis ticks and labelling, clear colorbars and instructive titles. On top of that, STAGLAB offers two plotting modes, **Analysis Mode** (`SWITCH.AnalysisMode`) to carefully examine the data and **Publication Mode** (set as default) to clearly present the data (Fig. 9). *Analysis Mode* offers detailed information and has refined axis ticks and more labels. *Publication Mode* shifts the focus from the information surrounding the data to the data itself and labels subplots automatically to be easily referred to. This is achieved by removing unnecessary (e.g., duplicated) labelling, the choice of a well-readable and larger sans-serif font and a less distracting grey colouring (see Sect. 3.2).

STAGLAB makes plotting multiple subplots straightforward, while keeping the figure clear and focussed. To compare different experiments, STAGLAB fully automatically adds subtle **visual background guides** (`SWITCH.BackgroundGuides`), areas that visually combine subplots of the same experiment while separating them visually from the other experiments (e.g., 20  Fig. 10). A similar examples is the **Time-Evolution Mode** (`SWITCH.TimeEvolutionMode`) that adds time arrows to highlight the temporal evolution from one subplot to another (Fig. 6). To highlight smaller areas in a subplot and enlarge important details, **magnifier panels** (`SWITCH.Magnifier`) can be added to plots (e.g., Fig. 7c). STAGLAB has a convenient option to use a discrete colour map (`SWITCH.DescreteColormap`) that can help to outline regions of similar values more clearly (Fig. 8). Finally, a stunning **Dark Mode** (`SWITCH.ColorMode`) can be switched on, which inverts relevant colours of the 25  figure to be presented on a black background (Fig. 7). *Dark Mode* is particularly useful to display STAGLAB figures on screens and via projectors. It is worth mentioning that all of these options and modes can be individually put into action with one single, simple switch. Moreover, options like these crucially enable STAGLAB figures to convey scientific data accurately and clearly, while still being visually appealing to the reader.

### 4.2.4  Plot types

30  Apart from plotting various parameter fields, STAGLAB can further produce various useful spatial or temporal graph plots (with data extracted from parameter fields) and a suite of special plots. The graph plots either show the **horizontal plate velocity** (`PLOT.PlateVelocity`; only for 2-D) or a **field-contour topography** (`PLOT.PlateBaseTopography`; e.g., for the lithosphere–asthenosphere boundary). Additionally, **StagLab-data** graphs (`PLOT.CustomGraph`) can be used to visualise a





large variety of previously processed and saved Geodynamic diagnostics against each other or time (e.g., trench location, plate velocities, etc.; see Sect. 4.2.5). Moreover, STAGLAB produces a variety of additional, useful plot types that are listed below:

1. **Grid** (`PLOT.Grid`): The grid can be plotted separately or as an addition to a parameter field. This is particularly useful to check physical features against grid resolution or to highlight the grid's spatial variation. The grid lines can be plotted in actual resolution or coarser, if the grid is too fine to be resolved with the given figure resolution.

2. **Tracers** (`PLOT.Tracers`): Individual tracer information, like the tracer position and type, can be visualised in a separate plot.

3. **Streamfunction** (`PLOT.Streamfunction`): There is an option to visualise the flow field in a separate plot or on top of any other field in the form of flow contours. This is a useful way to show the pattern and spatial extent of mantle flow cells.

4. **Streamlines** (`PLOT.Streamline`): Instantaneous stream lines can be plotted on top of another parameter field or as a separate plot to highlight the flow pattern of individual particles.

5. **Quiver** (`PLOT.Quiver`): The option to plot distributed velocity arrows on top of any parameter field adds the possibility to highlight the flow direction and relative strength. The amount and scaling of velocity arrows can be adjusted manually if needed.

6. **Surface-Field Variation** (`PLOT.SurfaceFieldVariation`): Any field variation across a horizontal surface can be plotted separately as a histogram plot (see Supplementary Fig. S1).

7. **Plate Sketch** (`PLOT.PlateSketch`): STAGLAB has a useful option to draw a simplified sketch of the surface plates to a separate plot. The sketch clearly highlights the position of the plate boundaries (i.e., subduction trench and spreading ridge) and plots numbers indicating plate, trench and convergence velocities and if specified other diagnostics like lithospheric bending dissipation or slab dip.

8. **Parameter Table** (`PLOT.ParameterTable`): The possibility to add a table to the figure is helpful to highlight a specified selection of the numerous diagnostic variables obtained in STAGLAB.

### 4.2.5 Output files

STAGLAB produces **publication-ready figure files** in a variety of data formats. The available options include *.jpg*, *.png*, *.eps* and *.pdf* file formats, whereas the *.png* format is most recommended and hence the default. High-resolution *.png* files are a good option for publication because .png is a commonly used and accepted figure file format and has a relatively small file size. True vector graphics such as *.eps* are limited to simple graph plots as contour plots would lead to excessively large file size. The resolution of an output figure file can be adjusted in the parameter file, if necessary.

STAGLAB produces **publication-ready movie files** in a variety of data formats. Movies of for example time-dependent numerical models are more and more published alongside the publication manuscript as most scientific journals offer the




option to add online supplementary files. Movies are particularly helpful to investigate the temporal behaviour of a system. STAGLAB can therefore also produce movie files created of multiple MatLab figure frames. Available file formats are $.avi$, $.mj2$, $.mp4$ and $.m4v$.

STAGLAB produces **post-processed data files** in a variety of data formats. This might be useful to save post-processed

parameter field or simply to convert a field from one format to another. Available file formats are here $.mat$, $.dat$ and $.txt$. Particularly useful is the option to save a large variety of Geodynamic diagnostics to data files. The diagnostics that can currently be saved are listed in Supplementary Table S1. Having these data files is particularly useful to plot temporal (or other) graphs including some of the diagnostic data using the option to plot this STAGLAB data (see Sect. 4.2.4).

### 4.3 Software design

STAGLAB is written in MatLab and compatible with all the latest MatLab versions including MatLab 8.4.0 (2014b) and newer (see Sect. 4.4.1 for more details). STAGLAB has its roots in STAGPLOT, a plotting routine introduced in Crameri (2013), and has been developed and extended further ever since with a few externally contributed routines. It now consists of an incredible flexible parameter file that executes one of the three core applications STAGPLOT, STAGRPROF or STAGTIMEDAT. While STAGPLOT mainly handles two- or three-dimensional data of field variables, STAGRPROF and STAGTIMEDAT visualise data

of radial profiles of horizontally averaged variables (e.g., Supplementary Fig. S3) and time-evolution graphs of globally averaged variables (e.g., Supplementary Fig. S4), respectively. The parameter file allows to change chosen parameters and switches to be different from the default setting. As such, a STAGLAB procedure is fully reproducible by saving a used parameter file – even after updates to the core routines as the parameter files in STAGLAB are forward compatible. Moreover, STAGLAB is extensively tested and heavily optimised for efficiency, which speeds up both computation and, crucially, also the research

process as a whole (see Sect. 4.3.5). It also makes elaborated diagnostics and state-of-the-art visualisation easily accessible to everyone. And, last but not least, the Geodynamic post-processing routines are fully open source.

#### 4.3.1 External contributions to StagLab 3.0

STAGLAB calls the routine $f\_readStagYY.m$ that was originally written by Boris Kaus to read StagYY's binary output directly into *MatLab*. The routine $f\_YYtoMap$, which was originally written by Paul Tackley, is used to produce horizontal

maps of fully spherical yinyang data. The original routine $f\_readFluidity$ to read Fluidity data was provided by Fanny Garel. It further uses the figure saving routine $export\_fig$, which was originally written by Oliver Woodford, the routine $flowfun$ originally written by Kirill K. Pankratov to derive the stream function, the routine $MinVolEllipse$ by Nima Moshtagh to fit a minimum-volume ellipse around a point cloud, the routine $plotboxpos$ by Kelly Kearney to derive the plot position more accurately, and the routine $hatchfill2$ originally developed by Neil Tandon to fill areas with a specific texture.



### 4.3.2 Flexibility

STAGLAB is built for flexibility. Its fully customisable parameter files contain a wealth of options (i.e., switches) to adjust the post-processing and visualisation. Data from all model geometries, ranging from simple 2-D Cartesian to 3-D fully spherical, can be processed (see Sect. 4.2.4). Various plot additions like velocity arrows, stream lines or isolines can be added on top of other parameter fields (see Sect. 4.2.4). An automatic subplot arrangement enables direct comparisons between outputs from different time steps or even experiments.

### 4.3.3 Reproducibility

STAGLAB is built for reproducibility. The customised parameter files, from which any STAGLAB procedure is executed, are forward compatible and can be stored in and run from any possible directory. Given that a previously used parameter file is safely stored, it can therefore be reused at any time, with any forthcoming version of STAGLAB and so, reproduce previous Geodynamic diagnostics and visualisations. Therefore, any work done with STAGLAB is, and always will be, fully reproducible.

### 4.3.4 Continuity

STAGLAB is build for continuity. Old parameter files are, on the one side, always updated to be compatible with the latest version of STAGLAB, fully automatically and fully effortlessly, while, on the other side, STAGLAB itself will be kept compatible with the latest versions of MatLab.

### 4.3.5 Efficiency

STAGLAB is build for efficiency. A wealth of most common post-processing and visualisation tasks are just a click (and a few seconds or less) away. Crucially, STAGLAB diagnostic and visualisation tasks can be quickly reproduced when going through the common, improving iterations during the research progress. This keeps the time-consuming coding effort for producing, adjusting and updating the core software to a minimum.

In addition, STAGLAB itself is trimmed towards efficient computation. Time-consuming parts of the software are optimised by, for example, vectorisation of loops as well as efficient reading and transferring of large data structures. It can thus quickly handle the huge amount of data resulting from high-resolution 2-D and 3-D models. A useful performance switch is built in that allows to reduce the file sizes of extremely large data files for quicker visualisation (`SWITCH.ReduceSize`). Another option, **Quick Mode** (`SWITCH.QuickMode`), allows for an extra quick execution to ensure efficient post-processing throughout multiple time steps and files.

### 4.3.6 Reliability

STAGLAB is build for reliability. It has been coded carefully and tested extensively to prevent foreseeable problems. Although it is always optimised for the latest MatLab version, STAGLAB's compatibility with previous MatLab releases is also tested





carefully. It is constantly being debugged and tested, and a growing number of users accelerates the exposure of hidden problems. Bug reports are very welcome and should be send to the author to ensure future reliable releases of STAGLAB. However, it has to be noted that since a fully bug-free software cannot be guaranteed, the scientific quality check remains always with the user.

### 4.3.7 Simplicity

The user-interface (UI) design is a pivotal part of any software to simplify its application. STAGLAB uses therefore parameter files combining all the important switches in one place. The parameter file is clearly structured and can be reduced to only the routinely used switches thanks to having defaults to every individual switch. Using the parameter files ensures a streamlined usage even for complex figures. STAGLAB returns selected, clear and informative real-time output in MatLab's terminal window during its operations. The display messaging allows to monitor STAGLAB's progress, receive both scientific diagnostics and, in case of problems, clear warnings thanks to its advanced error messaging system. The advanced error handling implemented in STAGLAB often prevents interruption during execution and displays warnings instead. An optional **VerboseMode** (`SWITCH.Verbose`) allows to display more-detailed information during the STAGLAB execution, which simplifies the debugging procedure. STAGLAB crucially simplifies reading and saving of data files: An advanced file finder automatically checks for other possible file directories or the latest file number, if the specified files are not found initially. Finally, STAGLAB itself is written carefully with a unified code structure and descriptive comments to simplify further code development.

### 4.3.8 Open source

Apart from the design routines, all STAGLAB files are fully open source. Free usage and re-distribution of STAGLAB and its individual routines and colour schemes falls, however, under the Creative Commons Attribution 4.0 International License.

### 4.4 Using StagLab

STAGLAB's software design ensures an enjoyable user experience. Its application is simple and hence accessible to experienced numerical modellers as well as fresh beginners. Using STAGLAB is an incredible efficient and motivating way, for students especially, to enter the world of numerical modelling, but also opens up new exciting doors for the more experienced scientists through its incredible flexibility.

### 4.4.1 Prerequisites

STAGLAB is compatible with all the three major operating systems running on Mac, Linux, and Windows PC. It has been updated to the latest graphical improvements made to MatLab and therefore performs best with the latest MatLab version. Although it can function with older versions, MatLab version 8.4.0 (i.e., MatLab 2014b) or higher is necessary for many core functions and thus highly recommended.

(c) Author(s) 2018. CC BY 4.0 License.





Although a set of parameter-field data is necessary to use all the post-processing routines build into STAGLAB, it can already function with just a temperature field, or any other single field that has to be processed. To complete more demanding tasks like plate-boundary tracking, more fields like temperature, velocity and topography are necessary.

STAGLAB performs a local directory search for the specified input file in case the file cannot be found directly. It also checks

for the specific folder structure (based on StagYY's native folder structure) consisting of an image folder '.../<folder>/+im' containing the image files and an output folder '.../<folder>/+op' containing the (e.g., binary) output data files. Using default settings, STAGLAB will, in that case, look for binary data to import in '+op', while it will save the figures to '+im'. This behaviour can, however, also be adjusted manually in the parameter file.

### 4.4.2   Download and installation

STAGLAB is available from www.fabiocrameri.ch/software. An included README file provides up-to-date, detailed instruction on installing and running STAGLAB. Once downloaded, it can then be installed by adding all of its files to the MatLab search path. This can conveniently be done by running the included installation routine, $f\_INSTALL$ (Algorithm 1), which additionally checks for possible file duplications, or manually from within MatLab (i.e., version 2014b or later).

---

**Algorithm 1** Installing command for StagLab

---

```
>> cd <yourPath>/StagLab3
>> f_INSTALL
```

---

### 4.4.3   Testing

STAGLAB has a build-in testing routine, $f\_TEST$ (Algorithm 2), to make sure it performs as expected on the current system and to highlight some of its capabilities. It performs a suite of automated tests for STAGLAB's core tasks and produces a suite of test figures from some included data files.

---

**Algorithm 2** Testing command for StagLab

---

```
>> cd <yourPath>/StagLab3
>> f_TEST
```

---

### 4.4.4   Running StagLab

Once it has been added to the MatLab search path, STAGLAB can be run via one of the provided parameter files (or a copy

thereof). Included example parameter files are $ParStagLab2D.m$ and $ParStagLab3D.m$ for Cartesian two- and three-dimensional models, respectively, and $ParStagLabYY.m$ for spherical models using the yinyang grid. The $ParStagLabRprof.m$



and $ParStagLabTimedat.m$ are two additional example parameter files to visualise either preprocessed radial profiles of horizontal mean values or the preprocessed temporal evolution of global mean values. The parameter files can conveniently also only contain a few specific switches as the full set of options is included in one of the three corresponding files containing the full default setups, $f\_Defaults.m$, $f\_DefaultsRprof.m$, or $f\_DefaultsTimedat.m$.

Old parameter files are automatically updated to be forward-compatible with any upcoming STAGLAB version and are hence fully reusable. This ensures that any figure produced with one specific parameter file remains fully reproducible given that a copy of the used parameter file is stored safely.

The simple STAGLAB procedure is then as follows:

0. **Downloading and installing STAGLAB** by adding all included files to the MatLab search path (e.g., running $f\_INSTALL$).

1. **Setting up new STAGLAB parameter file** with the output-data specific settings like file name, directory, number and parameter setup.

2. **Executing STAGLAB parameter file** with the user-specific switches to get publication-ready figures or movies.

3. **Safe-keeping STAGLAB parameter file** to ensure reproducibility.

### 4.4.5 Application examples

STAGLAB's parameter files can consist of only a minimum number of switches despite the numerous potential switches and options. The minimal parameter file to produce a figure similar to Fig. 7 is outlined in Algorithm 3.

STAGLAB diagnostics and visualisations can be used in two ways. First, they give useful insights during the testing of potential models and setups. Running STAGLAB in *Analysis Mode* produces, for example, more detailed plots with more information.

Secondly, STAGLAB diagnostics and visualisations can be used to present and publish new scientific results in a clear and appealing manner. STAGLAB has already been used in a number of studies already (Crameri, 2013; Crameri and Tackley, 2014, 2015, 2016; Crameri et al., 2017). Its fully automated diagnostics were pivotal for Crameri et al. (2017), where it enabled a to-date unmatched extensive, systematic testing of the numerous controlling subduction parameters on their impact on surface topography. Its visual representation of plate diagnostics even unravelled the dramatic interaction of subduction-induced mantle

currents and (upper-) plate tilting during the short time interval when the sinking plate reaches the lower mantle (see Crameri and Lithgow-Bertelloni, 2017), a dynamic interaction that has been overlooked in numerous other similar studies for years.

### 5 Conclusions

Firstly, a compilation of automised key plate-tectonics, mantle-dynamics and field-variation diagnostics is given here. Most of these diagnostics have been previously established elsewhere, some of these were improved here or even newly introduced.

Covered are, amongst others, surface-topography component splitting, plate-boundary identification, slab-polarity recognition,





---

**Algorithm 3** Minimal parameter file example

---

```
%% INPUT FILE(S)
IN.Name            =   { 'Case1' };
IN.Number          =   [ 20 ];
IN.Parameter       =   [ 1 ];
IN.Folder          =   { '../' };

%% POST-PROCESSING
PLOT.indicateTrench   =   logical(1);
PLOT.indicateRidge    =   logical(1);
PLOT.indicateBending  =   logical(1);
PLOT.indicateSlabTip  =   logical(1);

%% PLOT STYLING
STYLE.ColorMode       =   'dark';

%% PLOT ADDITIONS
PLOT.Magnifier     =   logical(1);

%% SAVING FIGURE
SAVE.Figure        =   logical(1);

%% FIELDS TO PLOT
PLOT.Temperature   =   logical(0);
PLOT.Viscosity     =   logical(1);
PLOT.Topography    =   logical(1);

%% SPECIAL PLOTS
PLOT.PlateSketch   =   logical(1);
```

---

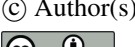


plate-bending characteristics derivation, and mantle-plume detection. They serve to offer a better understanding of complex Geodynamic systems like the Earth's mantle.

Secondly, the importance of scientific visualisation is highlighted for the communication of precious scientific ideas and results. To enable figures that live up to todays gold standard of scientific visualisation, a novel set of scientifically tested colour
schemes is introduced here. These novel, perceptually uniform colour maps are designed to prevent common scientific pitfalls and eradicate unnecessary visual errors, which otherwise can mount up to a staggering 7.5 %, as is shown to be the case for the commonly used *rainbow* colour map; An error that easily dominates in most data sets. The novel suite of scientific colour maps, including '*devon*', '*davos*', '*oslo*', '*broc*' and others, presents the precious scientific data undistorted and without excluding certain readers. In an unprecedented manner, the whole suite is made freely available (see www.fabiocrameri.ch/visualisation).
Thirdly, all Geodynamic diagnostics and the state-of-the-art scientific visualisation is packed into one single software package, STAGLAB, which is introduced here. STAGLAB is an easy-to-use MatLab software that significantly facilitates post-processing and visualisation, two important aspects of research. It provides a powerful, fully automated and incredibly robust implementation of the Geodynamic diagnostics outlined above. STAGLAB's efficiency turns laborious days of post-processing towards the hidden secrets of a model into exciting and effortless 2.2 seconds of pure revelation (according to measurement
outlined in Sect. 4.2.2). In the same breath, these revelations can be finely packed into a publication-ready figure or movie using fully-reproducible, forward-compatible parameter files, while applying state-of-the-art visualisation techniques like its unique suite of scientifically-tested, perceptually-uniform colour maps.

STAGLAB is currently compatible with two of the widely used Geodynamic codes, StagYY (Tackley, 2008) and Fluidity (Davies et al., 2011). In combination with StagYY output, it is capable of handling all different geometries (2-D and 3-D
Cartesian, 2-D partial and full cylindrical, and 3-D spherical) and output (parameter fields, radial profiles, and time evolution data). With little effort, STAGLAB can also be adjusted to be compatible with additional mantle convection codes. The latest version of STAGLAB, freely available at www.fabiocrameri.ch/software, is a flexible, efficient, reliable and simple software that produces state-of-the-art, reproducible diagnostics and visualisation for upcoming and groundbreaking Geodynamic models.

*Code and data availability.*   The full suite of scientific-proof colour maps can be freely downloaded from www.fabiocrameri.ch/visualisation.
The full STAGLAB package can be freely downloaded at www.fabiocrameri.ch/software with example data sets included.

*Competing interests.*   The author declares that he has no conflict of interest.

*Acknowledgements.*   The author thanks Kiran Chotalia and Antoniette Grima for their help with debugging the code, Fanny Garel for sharing her post-processing routines for Fluidity output, Robert Petersen for his help with deriving an appropriate radius of curvature for the plate bending, Tobias Rolf for his help with the histogram plot, Marcel Thielmann for his help with STAGLAB's compatibility across different





MatLab versions, and Paul Tackley for his helpful comments throughout the development of STAGLAB. The author acknowledges support from the Research Council of Norway through its Centers of Excellence funding scheme, Project Number 223272. Some of the models presented here were computed using a UNINETT Sigma2 computational resource allocation (Notur NN9283K and NorStore NS9029K).



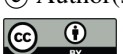

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



**Table 1.** Regional topographic characteristics

| Geometric characteristic | Abbreviation | Measurement method [a,b,c] |
|---|---|---|
| **Viscous fore-bulge** | $FB$ | $FB = FB_0 - z_2$ |
| **Subduction trench** | $TR$ | $TR = TR_0$ |
| **Island arc** | $IA$ | $IA = IA_0 - z_4$ |
| **Back-arc depression depth** | $BAD$ | $BAD = BAD_0 - z_4$ |
| **Back-arc depression extent** | $BAD_{extent}$ | $BAD_{extent} = x_{TR} - x_3$ |
| **Back-arc depression area** | $BAD_{area}$ | $BAD_{area} = \int_{x_3}^{x_4} z_3 - z(x)\,\mathrm{d}x$ |

[a] $x_1 = 0$, $x_2 = 400$ km, $x_3 = x(z = z_3)$, and $x_4 = x_{TR}$.

[b] $z_1 = 0$ km is the sea level, $z_2 = z_{min}(x > x_{FB})$, $z_4 = z_{mean}(x < x_2)$, and $z_3 = z_4 - 2(z_{max} - z_{min})$ for $x < x_2$.

[c] see graphical representation in Fig. 2.



**Table 2.** STAGLAB's main 2-D Geodynamic diagnostics

| Diagnostics | Availability[a] | | |
| --- | --- | --- | --- |
| | 2-D Cartesian | 2-D cylindrical | 3-D Cartesian (*2-D mode*[b]) |
| *Topography* | | | |
| Regional characteristics | ✓ | ✓ | ✓ |
| Isostatic topography component | ✓ | ✓ | ✓ |
| Residual topography component | ✓ | ✓ | ✓ |
| *Plate* | | | |
| Convergent-boundary tracking | ✓ | ✓ | ✓ |
| Divergent-boundary tracking | ✓ | ✓ | ✓ |
| Plate thickness | ✓ | ✓ | ✓ |
| Plate-core stress | ✓ | ✓ | ✓ |
| Plate-core strainrate | ✓ | ✓ | ✓ |
| Max. depth of plastic failure | ✓ | ✓ | ✓ |
| Subduction kinematics | ✓ | ✓ | ✓ |
| Subduction polarity | ✓ | ✓ | ✓ |
| Subducting-plate age at trench | ✓ | ✓ | ✓ |
| Subducting-plate bending | ✓ | ✓ | ✓ |
| Subduction flow rate | ✓ | ✓ | ✓ |
| Plate bending dissipation | ✓ | ✓ | ✓ |
| Overriding-plate tilt | ✓ | ✓ | ✓ |
| Spreading kinematics | ✓ | ✓ | ✓ |
| *Slab* | | | |
| Slab viscosity | ✓ | ✓ | ✓ |
| Slab-mantle viscosity contrast | ✓ | ✓ | ✓ |
| Slab-tip depth | ✓ | ✓ | ✓ |
| Shallow-depth slab dip angle | ✓ | ✓ | ✓ |
| Slab-sinking velocity | ✓ | ✓ | ✓ |
| Slab water retention | ✓ | ✓ | ✓ |
| *Mantle* | | | |
| Mantle transit time | ✓ | ✓ | ✓ |
| Upper-mantle viscosity | ✓ | ✓ | ✓ |
| Mantle-plume tracking | ✓ | ✓ | ✓ |
| Active- vs. passive up-/downwelling | ✓ | ✓ | ✓ |
| Total up-/downwelling volume | ✓ | ✓ | ✓ |

[a] At time of submission. [b] Diagnostics performed on a vertical cross-section through a 3-D model.



**Table 3.** Necessary fields for STAGLAB's core geodynamic diagnostics$^a$

| Diagnostics | Necessary parameter fields$^a$ | | | | | | | |
|---|---|---|---|---|---|---|---|---|
| | Temperature | Velocity | Density | Viscosity | Composition | Stress | Strain rate | Topography |
| Topography components | ■ | | □ | | | | | ■ |
| Plat-boundary tracking | ■ | ■ | | ■ | | | | |
| Plate viscous dissipation | ■ | | | ■ | | ■ | ■ | |
| Subducting-plate age at trench | ■ | ■ | | ■ | | | | |
| Plate bending dissipation | ■ | ■ | | ■ | □ | | | |
| Slab tracking | ■ | | | ■ | | | | |
| Slab water retention | ■ | ■ | | ■ | | | | |
| Mantle-plume tracking | ■ | ■ | | | | | | |
| Active- vs. passive up-/downwelling | ■ | ■ | | | | | | |

■ Necessity for diagnostics. □ Improves the diagnostics without being a necessity. $^a$At time of submission.



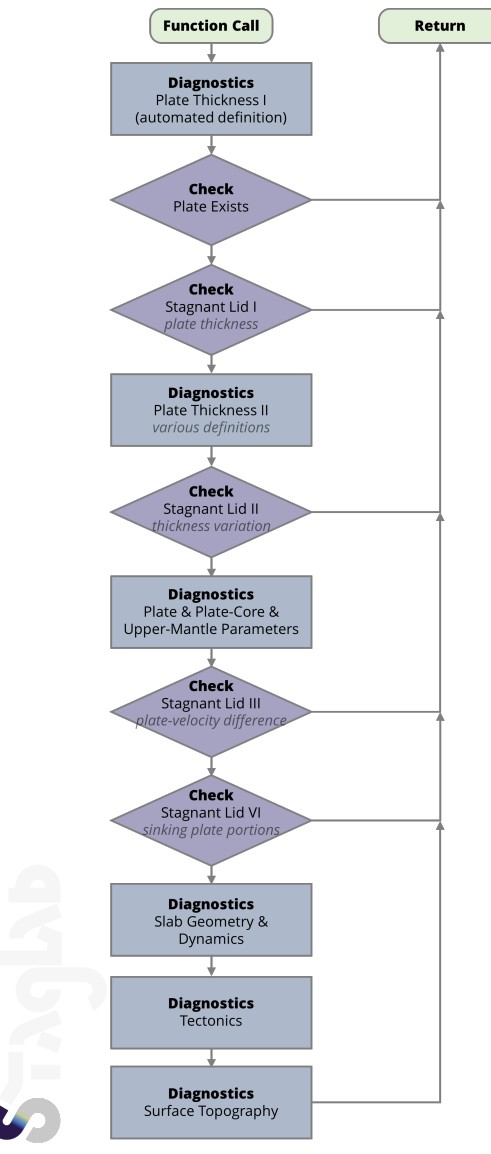

**Figure 1.** Flow chart outlining STAGLAB's plate-tectonics diagnostic procedure.





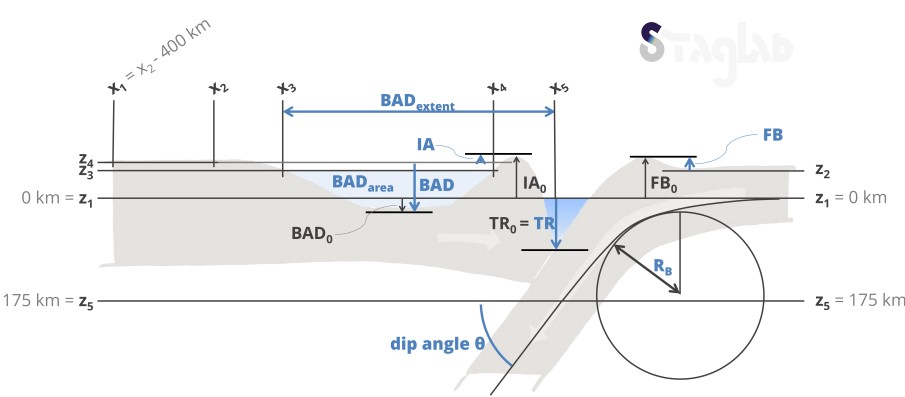

**Figure 2.** Deriving the regional topographic characteristics at a subduction zone, which are from the subducting plate (right) towards the overriding plate (left) viscous fore-bulge (FB), subduction trench (TR), island arc (IA), and back-arc depression (BAD). Additional diagnostics explained here are the shallow-depth slab dip, $\theta$, taken at the depth $z_5$, and the bending radius $R_B$ is derived by fitting a circle to the point of maximal plate bending. See text and Table 1 for more details (Figure reproduced from Crameri et al., 2017).





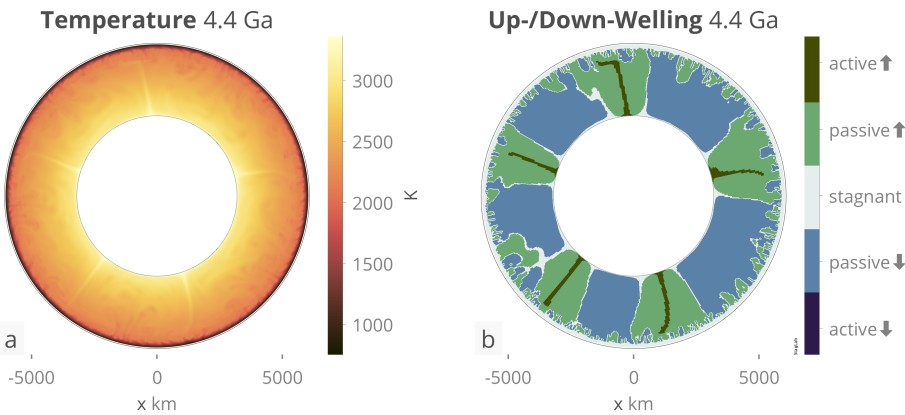

**Figure 3.** STAGLAB's plots showing a mantle-convection model in 2-D cylindrical geometry for (a) temperature and (b) diagnosed active and passive up- and downwellings.



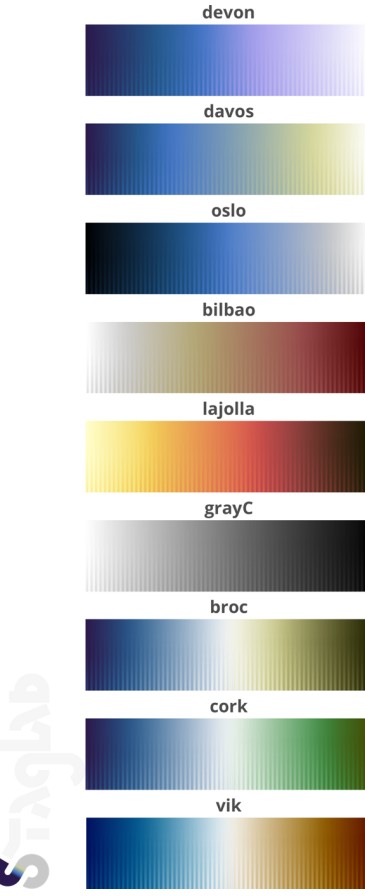

**Figure 4.** The novel set of scientific colour maps used in STAGLAB is freely available online (www.fabiocrameri.ch/visualisation) in all major data formats. The included colour maps, *devon*, *davos*, *oslo*, *bilbao*, *lajolla*, *grayC*, and *broc*, *cork* and *vik*, are all perceptually uniform and prevent distorting the data visually.




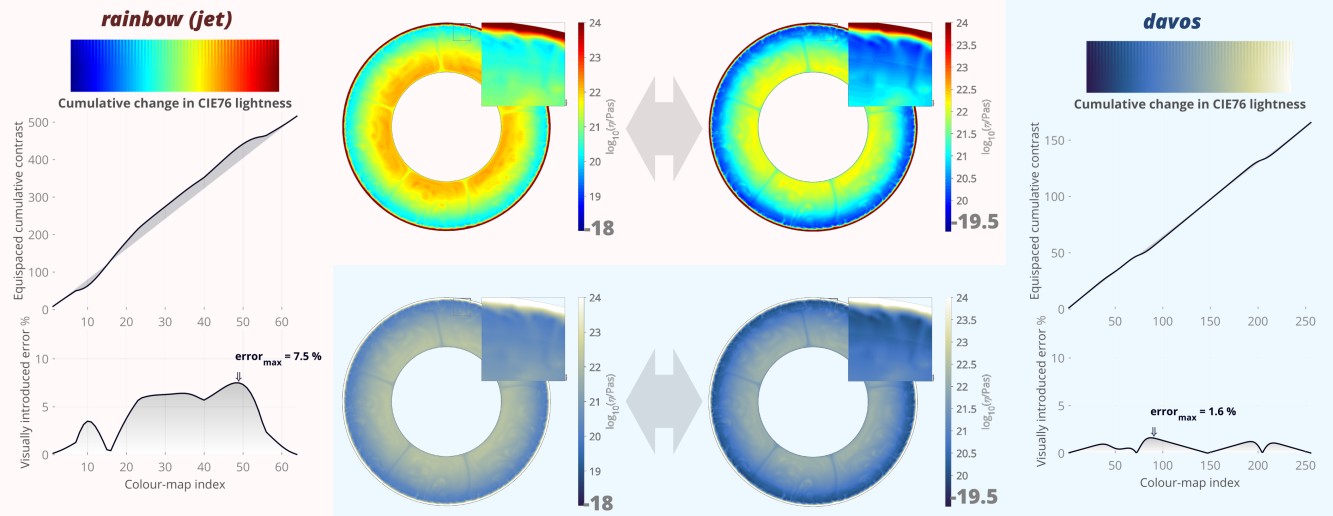

**Figure 5.** The visually introduced error to the underlying data. A common $rainbow$ (a.k.a. $jet$) colour map hides low-contrast variations in the cyan–green part and amplifies it unnaturally elsewhere as highlighted by low amplitude ripples in the two large colorbars (after Kovesi, 2015). CIE76-lightness variations along the standard $rainbow$ colorbar introduce a significant error of 7.5 % across the colorbar range to the underlying data, while the error of the perceptually uniform colour map $davos$ is only 1.6 % locally. The impact of the much higher visual error is dramatic and can be seen by a slight shift of colorbar limits: While the same data looks factually the same with the scientific $davos$ colour map, the unscientific $rainbow$ colour map introduces strong artificial boundaries and distortion to the underlying data.

.





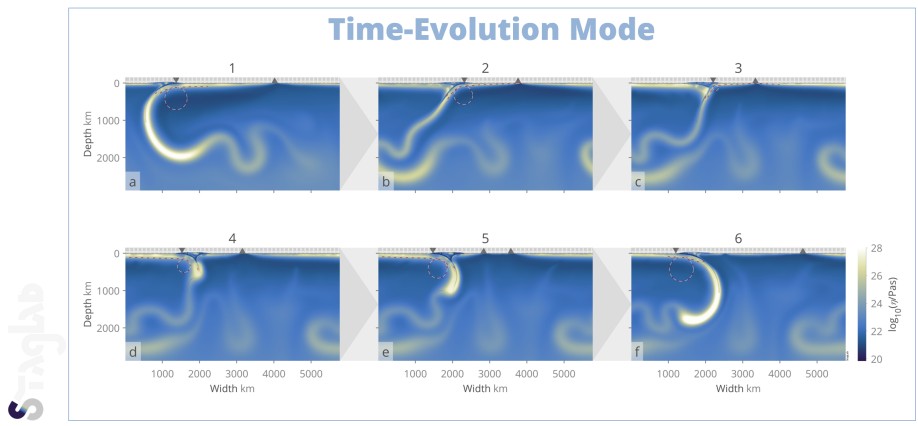

**Figure 6.** Time evolution of 2-D Cartesian mantle convection using STAGLAB's *Time Evolution Mode* highlighting the accuracy and robustness of the plate-boundary tracking algorithm even throughout a dramatic subduction-polarity reversal (Figure adjusted from Crameri and Tackley, 2015).





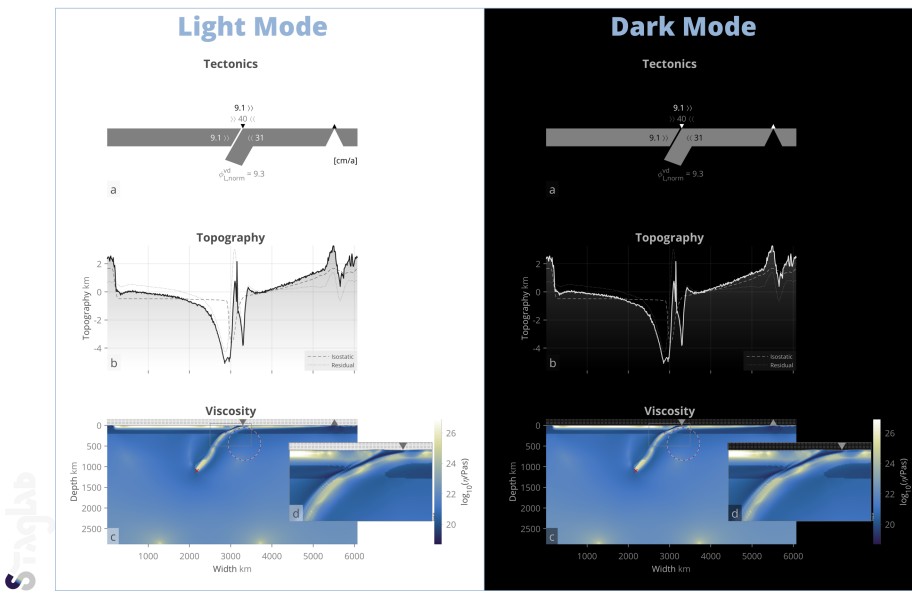

**Figure 7.** STAGLAB's *Dark Mode* and Geodynamic diagnostics. While the default *Light Mode* is intended for publications and poster presentations, *Dark Mode* is particularly useful for digital presentations on screens. STAGLAB's diagnostics highlighted here include isostatic and residual topography components (dashed and dotted grey lines, respectively), plate-boundary tracking (white & grey rectangles), and minimum plate-bending radius (red–white dashed circle) and resulting plate-bending dissipation foster better understanding of complex models.



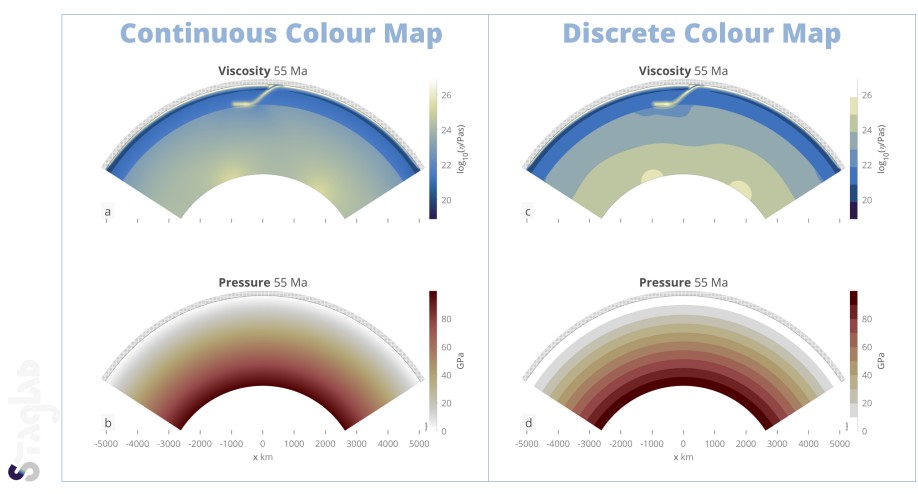

**Figure 8.** STAGLAB's continuous (a,b) versus its discrete (c,d) colour-map option shown for a model in partial-cylinder geometry.





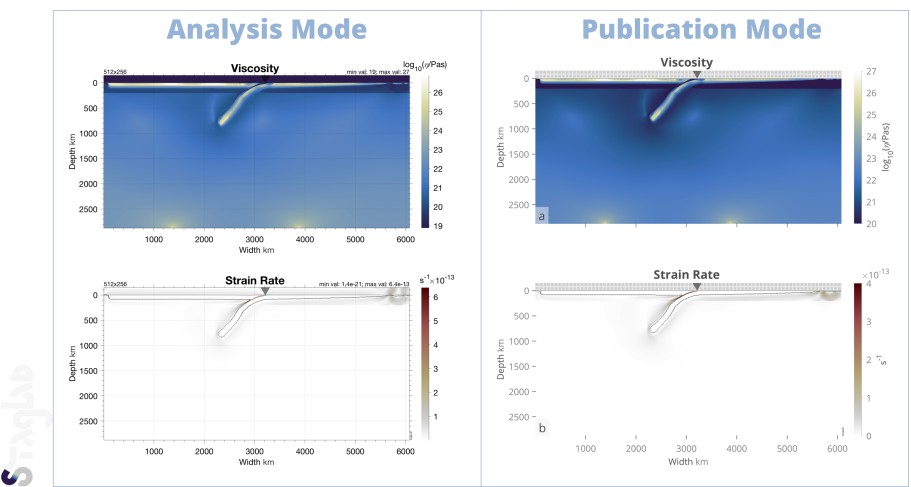

**Figure 9.** STAGLAB's *Analysis* (left) and *Publication Mode* (right). While in *Analysis Mode* as much detail as possible is provided to facilitate the examination of the data, the data (i.e., the key result) is put into focus and fully annotated in *Publication Mode*.



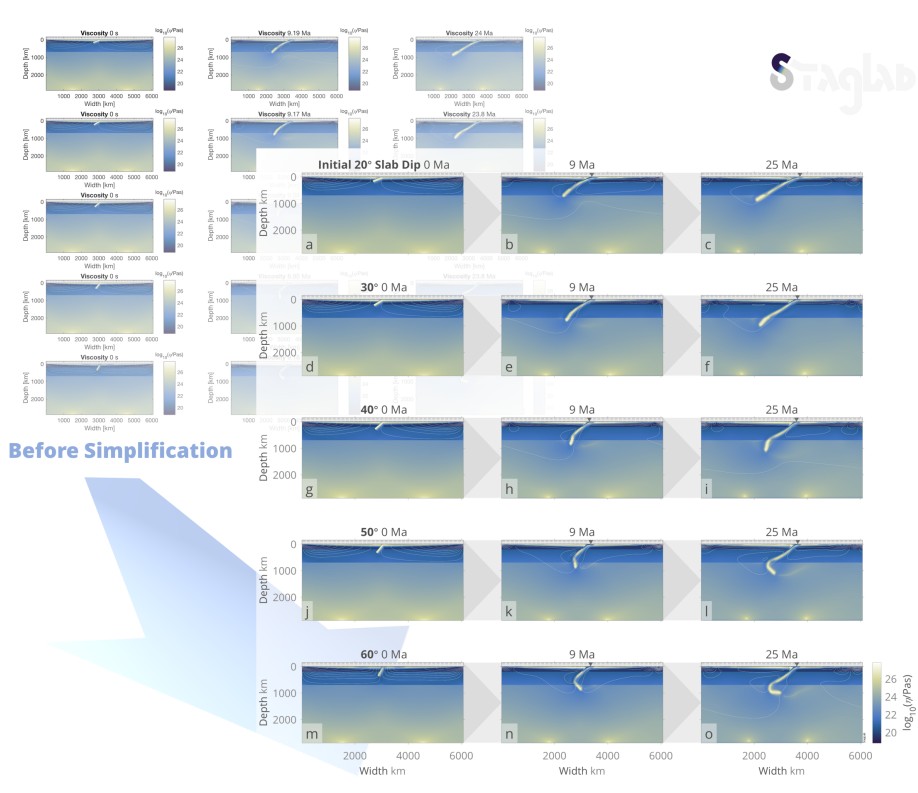

**Figure 10.** STAGLAB's clear and focussed plot design with various figure simplifications and subtle visual guides allows to compare a large number of models, or model time steps, while still keeping the figure easily understandable (Figure adjusted from Crameri and Lithgow-Bertelloni, 2017).