# Peer review of "Geodynamic diagnostics, scientific visualisation and StagLab 3.0"

_Geoscientific Model Development, 2017_

## Short Comment (SC1) · 15 Mar 2018

As explained in https://www.geoscientific-model-development.net/about/manuscript_types.html GMD does not see putting program code and data on author's own website as a mean for persistent access. Authors should consider improving the availability with a more permanent arrangement for instance to upload the code as a supplement or to provide a DOI to site providing permanent access, e.g Zenodo.

Lutz Gross GMD Executive Editor
* * *

---

## Author Comment (AC1) · 15 Mar 2018

StagLab 3 has been uploaded to Zenodo and been assigned a DOI number to ensure long-term availability:

doi:10.5281/zenodo.1199038

https://doi.org/10.5281/zenodo.1199038

A reference to this will be added to the manuscript.

---

## Referee Comment (RC1) · S. King (Referee) · 27 Mar 2018

1. Does the paper address relevant scientific modelling questions within the scope of GMD?

Yes. This is an important and timely contribution, while a number of open source codes are now provided to the community, there is a distinct lacking of analysis tools. Another more general comment, it is kind of odd to call this open source since it is built on MatLab. Indeed what you provide is open source but it only works with MatLab and someone who does not have access to MatLab will not be able to use it. You do mention this in the abstract but it's an important point. Finally, extending StagLab beyond output from StagYY is important if it will be adopted as a community tool, since

of course, StagYY is not available to many/most geodynammicists. One might find it ironic that a suite of open source analysis tools are being made available for a code that is not generally available. Yet this is a step in the right direction. Fluidity also has a fairly small user base. It appears that StagLab is extensible, so perhaps the challenge will be taken on by other users to write a function to import other formats. There seems to be a matvtk plugin for MatLab, I don't use MatLab anymore. I will note that vtk output is fairly standard and codes such as Citcom and Aspect support vtk output.

These are not a list of requirements to publish but things to think about. The idea of providing robust open-source analysis tools will greatly improve geodynamic modeling. Hopefully this will catch on and others will participate in the effort. One significant but time consuming step would be to translate this to python, which would free it from MatLab and make it really open source. It would be a significant effort but could be beneficial in the long run.

2. Does the paper present a model, advances in modelling science, or a modelling protocol that is suitable for addressing relevant scientific questions within the scope of EGU?

Yes. It is worth pointing out that the end the rainbow section is broader than geodynamic tools and this is an important contribution that could almost stand on it's own. It is relevant to all readers of GMD. I recall 25+ years ago researchers in visualization came to an AGU meeting showing how rainbow type color palettes distort perception. Ironically this session was juxtaposed with a session that introduced several new tomographic models were presented, of course using a rainbow-like (redinium/blutonium) palette. The references here are important and I'm glad to see this is once again a topic and I hope that with the sharing of tools and availability of these new palettes, I hope that others will adopt them.

3. Does the paper present novel concepts, ideas, tools, or data?

Yes.

4. Does the paper represent a sufficiently substantial advance in modelling science?

Yes.

5. Are the methods and assumptions valid and clearly outlined?

Yes.

6. Are the results sufficient to support the interpretations and conclusions?

Yes – not really relevant to this paper.

7. Is the description sufficiently complete and precise to allow their reproduction by fellow scientists (traceability of results)? In the case of model description papers, it should in theory be possible for an independent scientist to construct a model that, while not necessarily numerically identical, will produce scientifically equivalent results. Model development papers should be similarly reproducible. For MIP and benchmarking papers, it should be possible for the protocol to be precisely reproduced for an independent model. Descriptions of numerical advances should be precisely reproducible.

Yes.

8. Do the authors give proper credit to related work and clearly indicate their own new/original contribution?

Yes.

9. Does the title clearly reflect the contents of the paper? The model name and number should be included in papers that deal with only one model.

Yes

10. Does the abstract provide a concise and complete summary?

Yes.

11. Is the overall presentation well structured and clear?

Here there are some things to talk about. The output diagnostics begin with a list of specific diagnostics to subduction problems, which limits the usefulness of the tool. Beginning with more general output then moving to more specific cases would seem appropriate. In that sense starting with section 2.2 would seem more logical. What about Vrms and Nusselt number, and mean temperature as a function of depth? These all seem to be routine diagnostics of flow. As for plate-like flow, many people use plateness. Some of this seems to be embedded in sections 2.1.3 and 2.1.4 but they are more general than "plate tectonic diagonistics" and I would not expect to find stagnant lid diagnostics within a section labeled plate-tectonic diagnoistics. Organization of this is a bit scattered and does not help the interested reader. I would suggest something like: Generic flow diagnostics (Vrms, Nu, heat flow, dynamic topography, geoid?, mean temperature, depth average temperature and velocities); then plate/slab diagnostics, then plume tracking, etc.

12. Is the language fluent and precise?

One comment here. "STAGLAB's software design ensures an enjoyable user experience." Enjoyable? I'm not sure about this. How do you quantify that statement? After all what is enjoyable for one may not be for others. As the students in my geophysics class. I think you are trying to convey the idea that this is easy to use, flexible, and robust.

13. Are mathematical formulae, symbols, abbreviations, and units correctly defined and used?

Yes.

14. Should any parts of the paper (text, formulae, figures, tables) be clarified, reduced, combined, or eliminated?

no

15. Are the number and quality of references appropriate?

Yes.

Is the amount and quality of supplementary material appropriate? For model description papers, authors are strongly encouraged to submit supplementary material containing the model code and a user manual. For development, technical, and benchmarking papers, the submission of code to perform calculations described in the text is strongly encouraged.

This is fine.

---

## Referee Comment (RC2) · Anonymous Referee #2 · 2 Apr 2018

This manuscript addresses 3 different aspects of the post-processing of mantle convection numerical models. The first part could be described as a small review of some useful and robust diagnostics which can be used to describe the lithosphere and mantle behaviour of a complex and dynamic system. The second part points at the (well-known but often neglected!) problem of correctly displaying scientific results. More importantly, the last part presents StagLab 3.0, a code written in Matlab and capable of performing all the diagnostics described in the first two sections.

Scientific significance:

I read this manuscript with interest, since, as mentioned by the author, efficiently post-processing the outputs of mantle convection models is a growing concern in the geodynamics community and is time-consuming. Although this paper does not come with any

substantial new concept, it is an efficient and flexible open-source tool to post-process global and regional mantle convection models which can be useful to a large number of researchers of the mantle geodynamics community. I would therefore recommend its publication in GMD.

Here-below, I indicate potential minor suggestions to improve the clarity and goals of the proposed manuscript.

Scientific quality:

StagLab 3.0 proposes to describe key plate tectonics and mantle processes, which are still investigated amongst the geodynamics community. Therefore, the proposed diagnostics can be potentially used to lead to significant scientific results. It uses model data to robustly analyse a large variety of processes (age, velocities, topography, plate boundaries, active/passive up/downwellings. . .). It provides at least a first step to more complex potential diagnostics, which will hopefully be developed later, in a global effort of developing open-source post-processing routines.

Scientific reproductibility:

- Although primarily built for StagYY (which is not fully open-source), StagLab 3.0 aims at being potentially applied to post-process the outputs of every mantle convection code. It has already been tested with Fluidity, providing the routine f_readFluidity. The author provides the routine f_readOther to import raw data from other convection codes. However, no details are provided about how to adapt this last routine. Even if it may be hard to quantify precisely, is it possible to give more insight on the nature of the adaptations needed to use StagLab with another convection code? For example, a large number of geodynamicists use CitcomS and Aspect codes. Is a StagLab routine envisioned to be publicly accessible to read outputs from such codes?

- GMD strongly encourages the submission of the code or a user manual in the supplementary material. Although I think that it is not whether a user manual is available.

Presentation quality:

The manuscript is well written, clear and well organised. However, in section 2.1, entitled 'Plate tectonics diagnostics', I would have followed the order of the Fig. 1 flow chart for simplicity and clarity purposes. I would therefore advise to start with the global plate diagnostics (thickness, stagnant lid, boundaries, age, velocities, topography...) and finish with the more regional diagnostics (subduction topography, slab-dynamics and plate bending), which are more specific and less likely to be used for a large number of scientific purposes.

Technical corrections:

- P19, line 18: "StagLab is built" instead of "build" - Table 2 (third column) and first paragraph of section 2 (p19-20): is StagLab applicable to 2D slices of spherical models? It seems that it is the case but it's not clearly written. - Table 3, column 1: a 'e' is missing for Plate-boundary tracking - Figure 3: A stagnant-lid model is shown, therefore, no active subduction detected on Fig. 3b. Nevertheless, since the goal of this figure is to show that active/passive downwellings/upwellings are properly recovered by StagLab, I would have chosen a model displaying both plumes and slabs. - Fig. 6: Timestep 6: a ridge is not detected although the lithosphere is thin. Is it because the velocity threshold used to detect ridges is too high? - Fig. 7: What do the red crosses at the bottom of the slab mean? They are not described in the caption. Fig. 10: For simplicity, I would have just put the two sets of figures next to each other instead of adding perspective and transparency
* * *

---

## Author Comment (AC4) · 26 Apr 2018

I included all other minor suggestion by the reviewer.
* * *

---

## Author Comment (AC5) · 26 Apr 2018

see my answers to the similar comments of reviewer 1.

---

## Author Comment (AC6) · 26 Apr 2018

2-D slicing through 3-D spherical models is unfortunately not supported in the current version of StagLab, as this involves, in contrast to Cartesian models, computationally heavy interpolation of all grid data onto a new plane. I have clarified this in Table 2 and the text.

I will add a basic StagLab user guide to the online supplement.

I replaced the figure of the stagnant-lid model with a figure of a mobile-lid model that features both active up- and downwellings.

Figure 6: The former ridge (see timestep 5) is disappearing around timestep 6, where the ridge divergence velocity has already become lower than the detection threshold.

I added that red crosses explain slab-tip depth in the caption of Figure 7.

I included all other minor suggestions.
* * *

---

## Author Response (AR1)

**Review Response**

**I thank Scott King and an anonymous reviewer for their constructive thoughts and comments on the manuscript and underlying software.**

1. Does the paper address relevant scientific modelling questions within the scope of GMD?

Yes. This is an important and timely contribution, while a number of open source codes are now provided to the community, there is a distinct lacking of analysis tools. Another more general comment, it is kind of odd to call this open source since it is built on MatLab. Indeed what you provide is open source but it only works with MatLab and someone who does not have access to MatLab will not be able to use it. You do mention this in the abstract but it's an important point. Finally, extending StagLab beyond output from StagYY is important if it will be adopted as a community tool, since of course, StagYY is not available to many/most geodynamicists. One might find it ironic that a suite of open source analysis tools are being made available for a code that is not generally available. Yet this is a step in the right direction. Fluidity also has a fairly small user base. It appears that StagLab is extensible, so perhaps the challenge will be taken on by other users to write a function to import other formats. There seems to be a matvtk plugin for MatLab, I don't use MatLab anymore. I will note that vtk output is fairly standard and codes such as Citcom and Aspect support vtk output.

These are not a list of requirements to publish but things to think about. The idea of providing robust open-source analysis tools will greatly improve geodynamic modeling. Hopefully this will catch on and others will participate in the effort. One significant but time consuming step would be to translate this to python, which would free it from MatLab and make it really open source. It would be a significant effort but could be beneficial in the long run.

**It is true that StagLab is originally designed for StagYY and therefore currently works best with only that particular code. However, I put a lot of effort in it to open it up for potential use with other (especially open-source) codes. As mentioned by the reviewer, there is a specific file to try and guide through the adjustments needed for making StagLab compatible with other codes. Specific built-in error messages throughout StagLab then indicate necessary additional adjustments elsewhere. However, this is still not straightforward and needs my involvement, which is why I can currently not provide a more general guide to allow other developers to write their own conversion script. I will work on that and try to provide something along these lines in future versions of StagLab.**

**Most codes do unfortunately also not offer example output data, that would facilitate making StagLab compatible. However, I am now in contact with other developers to provide extended compatibility to other codes like ASPECT and CITCOM soon.**

**Regarding StagLab's native language MatLab, I generally agree that scientific codes should be accessible to anyone. However, I think one has to also consider the following aspects that seem to speak for MatLab and seem to favour effective user accessibility of MatLab over e.g., Python: The usage of MatLab is simpler than usage of e.g., Python; MatLab is probably**

**currently learned by more people than Python; most people that run Geodynamic models (often on a supercomputer) also have access to a MatLab license via their employers. On top of that, MatLab codes can, under some code simplifications, also be distributed as stand-alone applications that do not necessitate a licence for their execution. A light version of a stand-alone StagLab seems to be possible in the future.**

**I tried to clarify these points throughout the manuscript.**

2. Does the paper present a model, advances in modelling science, or a modelling protocol that is suitable for addressing relevant scientific questions within the scope of EGU?

Yes. It is worth pointing out that the end the rainbow section is broader than geodynamic tools and this is an important contribution that could almost stand on it's own. It is relevant to all readers of GMD. I recall 25+ years ago researchers in visualization came to an AGU meeting showing how rainbow type color palettes distort perception. Ironically this session was juxtaposed with a session that introduced several new tomographic models were presented, of course using a rainbow-like (redinium/blutonium) palette. The references here are important and I'm glad to see this is once again a topic and I hope that with the sharing of tools and availability of these new palettes, I hope that others will adopt them.

**I appreciate this comment highlighting the apparent need to revive awareness of the problems involved in unscientific visualisation amongst researchers, software designers and also editors.**

11. Is the overall presentation well structured and clear?

Here there are some things to talk about. The output diagnostics begin with a list of specific diagnostics to subduction problems, which limits the usefulness of the tool. Beginning with more general output then moving to more specific cases would seem appropriate. In that sense starting with section 2.2 would seem more logical. What about Vrms and Nusselt number, and mean temperature as a function of depth? These all seem to be routine diagnostics of flow. As for plate-like flow, many people use plateness. Some of this seems to be embedded in sections 2.1.3 and 2.1.4 but they are more general than "plate tectonic diagonistics" and I would not expect to find stagnant lid diagnostics within a section labeled plate-tectonic diagnoistics. Organization of this is a bit scattered and does not help the interested reader. I would suggest something like: Generic flow diagnostics (Vrms, Nu, heat flow, dynamic topography, geoid?, mean temperature, depth average temperature and velocities); then plate/slab diagnostics, then plume tracking, etc.

**Although it would be interesting, and probably useful to give a general overview over all important diagnostics applied in the field, this section is intended to cover only diagnostics that are included and performed in StagLab itself. I do therefore not include some basic diagnostics like the calculation of the geoid or the Nusselt number here in detail, but certainly consider adding these to a later version of the software.**

**However, visualising radial profiles or temporal graphs of globally averaged values of pre-calculated diagnostics (like Vrms, mean temperature, plateness and mobility) is built into StagLab (see e.g., Supplementary figure S3). I therefore added two sections explaining plateness and mobility. Again, I agree that calculating these kinds of radial profiles directly from the field data would be a useful addition to StagLab in the future.**

**I have now rearranged the order in this part of the manuscript and renamed the section headings to for clarification following the suggestions of the reviewer.**

12. Is the language fluent and precise?

One comment here. "STAGLAB's software design ensures an enjoyable user experience." Enjoyable? I'm not sure about this. How do you quantify that statement? After all what is enjoyable for one may not be for others. As the students in my geophysics class. I think you are trying to convey the idea that this is easy to use, flexible, and robust.

**I reworded the sentence removing the word 'enjoyable'.**

**Anonymous Referee #2**

This manuscript addresses 3 different aspects of the post-processing of mantle convection numerical models. The first part could be described as a small review of some useful and robust diagnostics which can be used to describe the lithosphere and mantle behaviour of a complex and dynamic system. The second part points at the (well- known but often neglected!) problem of correctly displaying scientific results. More importantly, the last part presents StagLab 3.0, a code written in Matlab and capable of performing all the diagnostics described in the first two sections.

Scientific significance:

I read this manuscript with interest, since, as mentioned by the author, efficiently post- processing the outputs of mantle convection models is a growing concern in the geodynamics community and is time-consuming. Although this paper does not come with any substantial new concept, it is an efficient and flexible open-source tool to post-process global and regional mantle convection models which can be useful to a large number of researchers of the mantle geodynamics community. I would therefore recommend its publication in GMD.

Here-below, I indicate potential minor suggestions to improve the clarity and goals of the proposed manuscript.

Scientific quality:

StagLab 3.0 proposes to describe key plate tectonics and mantle processes, which are still investigated amongst the geodynamics community. Therefore, the proposed diagnostics can be potentially used to lead to significant scientific results. It uses model data to robustly analyse a large variety of processes (age, velocities, topography, plate boundaries, active/passive up/downwellings. . .). It provides at least a first step to more complex potential diagnostics, which will hopefully be developed later, in a global effort of developing open-source post-processing routines.

Scientific reproducibility:

- Although primarily built for StagYY (which is not fully open-source), StagLab 3.0 aims at being potentially applied to post-process the outputs of every mantle convection code. It has already been tested with Fluidity, providing the routine f_readFluidity. The author provides the routine f_readOther to import raw data from other convection codes. However, no details are provided about how to adapt this last routine. Even if it may be hard to quantify precisely, is it possible to give more insight on the nature of the adaptations needed to use StagLab with another convection code? For example, a large number of geodynamicists use CitcomS and Aspect codes. Is a StagLab routine envisioned to be publicly accessible to read outputs from such codes?

**As mentioned by the reviewer, there is a specific file to try and guide through the adjustments needed for making StagLab compatible with other codes. Specific built-in error messages throughout StagLab then indicate necessary additional adjustments elsewhere. However, this is still not straightforward and needs my involvement, which is why I can currently not provide a more general guide to allow other developers to write their own conversion script. I will work on that and try to provide something along these lines in future versions of StagLab.**

**Most codes do unfortunately also not offer example output data, that would facilitate making StagLab compatible. I am, however, in contact with other developers to provide extended compatibility to other codes like ASPECT and CITCOM soon.**

- GMD strongly encourages the submission of the code or a user manual in the supplementary material. Although I think that it is not whether a user manual is available.

**I added a basic user guide to the online supplement.**

Presentation quality:

The manuscript is well written, clear and well organised. However, in section 2.1, entitled 'Plate tectonics diagnostics', I would have followed the order of the Fig. 1 flow chart for simplicity and clarity purposes. I would therefore advise to start with the global plate diagnostics (thickness, stagnant lid, boundaries, age, velocities, topography. . .) and finish with the more regional diagnostics (subduction topography, slab-dynamics and plate bending), which are more specific and less likely to be used for a large number of scientific purposes.

**I tried to clarify the structure of the mentioned sections by rearranging and renaming sections and subsections.**

Technical corrections:

- P19, line 18: "StagLab is built" instead of "build"

**fixed.**

- Table 2 (third column) and first paragraph of section 2 (p19-20): is StagLab applicable to 2D slices of spherical models? It seems that it is the case but it's not clearly written.

**2-D slicing through 3-D spherical models is unfortunately not supported in the current version of StagLab, as this involves, in contrast to Cartesian models, computationally heavy interpolation of all grid data onto a new plane. I have clarified this in Table 2 and the text.**

- Table 3, column 1: a 'e' is missing for Plate-boundary tracking

**fixed.**

- Figure 3: A stagnant-lid model is shown, therefore, no active subduction detected on Fig. 3b. Nevertheless, since the goal of this figure is to show that active/passive downwellings/ upwellings are properly recovered by StagLab, I would have chosen a model displaying both plumes and slabs.

**I replaced the stagnant lid model with a mobile lid model that features both active up- and downwellings.**

- Fig. 6: Timestep 6: a ridge is not detected although the lithosphere is thin. Is it because the velocity threshold used to detect ridges is too high?

**Yes, that is right: the former ridge (see timestep 5) is disappearing around timestep 6, where the ridge divergence velocity has already become lower than the detection threshold.**

- Fig. 7: What do the red crosses at the bottom of the slab mean? They are not described in the caption.

**I added that red crosses explain slab-tip depth in the caption.**

- Fig. 10: For simplicity, I would have just put the two sets of figures next to each other instead of adding perspective and transparency

**I agree and adjusted the figure accordingly.**

**Geodynamic diagnostics, scientific visualisation and StagLab 3.0**

Fabio Crameri

Centre for Earth Dynamics and Evolution (CEED), University of Oslo, Postbox 1028 Blindern, 0315 Oslo, Norway

*Correspondence to:* Fabio Crameri (fabio.crameri@geo.uio.no)

**Abstract.** Today's Geodynamic models can, often do, and sometimes have to become very complex. Their underlying, increasingly elaborate numerical codes produce a growing amount of raw data. Post-processing such data becomes therefore more and more important, but also more challenging and time consuming. In addition, visualising processed data and results has, in times of coloured figures and a wealth of half-scientific software, become one of the weakest pillars of science, widely mistreated and ignored. Efficient and automated Geodynamic diagnostics and sensible, scientific visualisation, preventing common pitfalls, is thus more important than ever. Here, a collection of numerous diagnostics for plate tectonics and mantle dynamics is provided and a case for truly scientific visualisation is made. Amongst other diagnostics are a most accurate and robust plate-boundary identification, slab-polarity recognition, plate-bending derivation, surface-topography component splitting and mantle-plume detection. Thanks to powerful image processing tools and other elaborate algorithms, these and many other insightful diagnostics are conveniently derived from only a subset of the most basic parameter fields. A brand-new set of scientifically proof, perceptually uniform colour maps including '$devon$', '$davos$', '$oslo$' and '$broc$' is introduced and made freely available (www.fabiocrameri.ch/colourmaps). These novel colour maps bring a significant advantage over misleading, non-scientific colour maps like '$rainbow$'$rainbow$', which is shown to introduce a visual error to the underlying data of up to 7.5 %. Finally, STAGLAB (www.fabiocrameri.ch/StagLab) is introduced, a software package that incorporates the whole suite of automated Geodynamic diagnostics and, on top of that, applies state-of-the-art, scientific visualisation to produce publication-ready figures and movies, all in a blink of an eye, all fully reproducible. STAGLAB, a simple, flexible, efficient and reliable tool, made freely available to everyone, is written in MatLab and adjustable for use with Geodynamic mantle-convection codes.

*Copyright statement.* StagLab, its individual subroutines and included scientific colour-map suite are licensed under a Creative Commons Attribution 4.0 International License.

[revised manuscript text omitted]